# Mutual Associations of Healthy Behaviours and Socioeconomic Status with Respiratory Diseases Mortality: A Large Prospective Cohort Study

**DOI:** 10.3390/nu15081872

**Published:** 2023-04-13

**Authors:** Min Du, Lin Zhu, Min Liu, Jue Liu

**Affiliations:** 1Department of Epidemiology and Biostatistics, School of Public Health, Peking University, No.38, Xueyuan Road, Haidian District, Beijing 100191, China; 2Center for Primary Care and Outcomes Research, School of Medicine, Center for Health Policy, Freeman Spogli Institute for International Studies, Stanford University, 450 Jane Stanford Way, Stanford, CA 94305-2004, USA; 3Institute for Global Health and Development, Peking University, No.5, Yiheyuan Road, Haidian District, Beijing 100871, China; 4Global Center for Infectious Disease and Policy Research & Global Health and Infectious Diseases Group, Peking University, No.38, Xueyuan Road, Haidian District, Beijing 100191, China; 5Key Laboratory of Reproductive Health, National Health and Family Planning Commission of the People’s Republic of China, No.38, Xueyuan Road, Haidian District, Beijing 100191, China

**Keywords:** socioeconomic status, healthy behaviours, respiratory diseases, mortality, cohort

## Abstract

Little cohort evidence is available on the effect of healthy behaviours and socioeconomic status (SES) on respiratory disease mortality. We included 372,845 participants from a UK biobank (2006–2021). SES was derived by latent class analysis. A healthy behaviours index was constructed. Participants were categorized into nine groups on the basis of combinations of them. The Cox proportional hazards model was used. There were 1447 deaths from respiratory diseases during 12.47 median follow-up years. The hazard ratios (HRs, 95% CIs) for the low SES (vs. high SES) and the four or five healthy behaviours (vs. no or one healthy behaviour) were 4.48 (3.45, 5.82) and 0.44 (0.36, 0.55), respectively. Participants with both low SES and no or one healthy behaviour had a higher risk of respiratory disease mortality (aHR = 8.32; 95% CI: 4.23, 16.35) compared with those in both high SES and four or five healthy behaviours groups. The joint associations were stronger in men than in women, and in younger than older adults. Low SES and less healthy behaviours were both associated with an increased risk of respiratory disease mortality, which augmented when both presented together, especially for young man.

## 1. Introduction

Across health systems, respiratory diseases have consistently brought a huge health burden [1]. Chronic respiratory diseases (CRD) became the third leading cause of death, behind cardiovascular diseases and neoplasms, in 2017 [2]. The World Health Organization (WHO) reported that chronic obstructive pulmonary disease (COPD) caused over three million deaths each year, accounting for nearly six percent of all deaths worldwide [3]. The United Kingdom (UK) had dramatic and high respiratory disease mortality from 1985 to 2015, especially infectious, interstitial, obstructive, and respiratory disease [4]. Therefore, preventing respiratory diseases, slowing their progression, and reducing the risk of death is vital in global health management.

Although socioeconomic progress and rising standards of living were witnessed in many countries in past decades, there was still an increasing rate of wealth inequity in the UK [5]. Moreover, the gap of socioeconomic inequity promoted the increased survival differences in the UK [5]. Unfortunately, low socioeconomic status (SES) resulted in reduction in life expectancy [6]. A systems review reported that three previous cross-sectional studies all found a non-significant relationship between chronic respiratory diseases and SES in low- and lower-middle-income countries, but one cohort study reported that higher SES (z-score method) decreased the risk of respiratory disease mortality [7,8]. In addition, a descriptive ecological study reported that for chronic respiratory disease mortality rate ratios, there were inequalities between the low and high strata using hierarchical classification [9]. Considering the above inconsistent results, different evaluation methods for SES, and lack of cohort studies, a large population-based cohort study was needed to analyse the relationship of SES with respiratory disease mortality, especially in high-income countries such as the UK with higher socioeconomic inequity [5].

Healthy behaviours are important and common factors influencing respiratory health and have extended people’s life span [10,11,12]. Recently, compared with a single healthy lifestyle, a healthy behaviours index that combined smoking status, alcohol consumption, physical activity, diet, and body mass index (BMI) came into view [13,14]. Several studies reported that the detrimental effect of low SES on the cardiovascular disease mortality could be alleviated by healthy behaviours [14,15]. However, its modification on the association of SES with respiratory disease mortality remains unknown. Moreover, studies on the joint effect of SES and healthy behaviours on respiratory disease mortality are lacking.

In this study, we aim to accomplish two goals by using UK Biobank cohort data with a large sample size. The primary goal is to investigate the impact of the association between SES and healthy behaviours on mortality from respiratory diseases, and the second goal is to explore whether findings are consistent among subpopulations by gender and age group.

## 2. Methods

### 2.1. Study Population

From 2006 to 2021, the UK Biobank (application 79114) enrolled 502,414 participants aged 37 to 73 years who resided within 40 km of 1 of 22 assessment centers across the United Kingdom (England, Wales, and Scotland) and were registered with the UK National Health Service (NHS). This large prospective cohort study covered multifarious and different settings, including urban–rural mix, socioeconomic, and ethnic heterogeneity. More details were shown in [16,17]. Among the 502,414 participants, after excluding those with missing information on socioeconomic status (n = 79,393), healthy behaviours (n = 33,967), and other covariates (n = 16,209), we finally included 372,845 participants (Figure 1). Appendix A presented missing information in detail. The National Information Governance Board for Health and Social Care and the NHS North West Multi-Centre Research Ethics Committee approved the UK Biobank study. Electronically signed consent has been obtained from all participants.

### 2.2. Assessment of SES

In our study, we assessed SES using total household income before tax, education qualifications, and employment status based on the previous study [15]. Total household income before tax was obtained through questionnaires, with options of <£18,000, £18,000–£30,999, £31,000–£51,999, £52,000–£100,000, >£100,000, do not know, or prefer not to answer. Education qualifications included eight options: college or university degree; advanced (A) levels, advanced subsidiary (AS) levels or equivalent; general certification of education ordinary (O) level, general certificate of secondary education (GCSEs) or equivalent; certificate of secondary education (CSEs) or equivalent; national vocational qualification (NVQ), higher national diploma (HND), higher national certificate (HNC), or equivalent; other professional qualifications; none of the above (equivalent to less than a high school diploma); or prefer not to answer. We regrouped employment status into two groups: unemployed and employed, which included those retired, in paid employment or self-employed, doing unpaid or voluntary work, or being full- or part-time students. Latent class analysis was used to create an overall SES variable which was identified as three latent classes, representing a low, medium, or high SES in terms of the item-response probabilities. Detailed information on latent class analysis is presented in the Methods section of the Appendix [15].

### 2.3. Assessment of Healthy Behaviours and Other Covariates

Five factors, including smoking status, alcohol intake, physical activity, diet scores, and body mass index (BMI, kg/m^2^) were used to construct a healthy behaviours index based on previous UK biobank studies (details are shown in Appendix A) [13,14,18]. Alcohol intake considered participants’ self–reported intake of six classes of alcoholic drinks per week and month. Moderate alcohol intake was viewed as five to 15 g of alcohol per day for women and five to 30 g per day for men [14]. The total metabolic equivalent task (MET) minutes per week for all activities, including walking, moderate, and vigorous activity was calculated based on the International Physical Activity Questionnaire [19]. Either equal or more than 735 MET min/week was considered sufficient physical activity [14]. Diet scores (theoretical range: zero to seven) were constructed based on six main food groups from a touchscreen food frequency questionnaire, in which values greater than three were viewed as a healthy diet [14,20]. For each behavioural component, we assigned one point for a healthy level (zero points for an unhealthy level), then added up the points that ranged between zero and five, and classified the healthy behaviour index into three groups (no or one, two or three, and four or five).

Other covariates, including age, gender, race and ethnicity, general health, weight loss, cancer, cardiovascular disease, diabetes, family history of diseases (high blood pressure, diabetes type 2, stroke, chronic bronchitis/emphysema, lung cancer, bowel cancer, breast cancer, Parkinson’s disease, Alzheimer’s disease/dementia, and severe depression), poor psychological status, sleep duration, and coffee and tea intake, were investigated through questionnaires. We classified adults’ and older adults’ sleep duration into three groups: normal sleep duration, short sleep duration, and long sleep duration, based on recommendations from the National Sleep Foundation [21].

### 2.4. Assessment of Respiratory Disease Mortality

Outcomes, including vital status, date of death, and underlying primary cause of death by 30 June 2020, were provided by the NHS Information Centre (England and Wales) and the NHS Central Register (Scotland) [22]. Deaths from respiratory diseases were defined according to the codes from the International Statistical Classification of Diseases and Related Health Problems, 10th Revision (ICD-10): J00-J99 total respiratory diseases; J09–J18 influenza and pneumonia; J40–J47 chronic lower respiratory diseases [23].

### 2.5. Statistical Analysis

We presented baseline characteristics as mean (standard deviation, SD) or median (interquartile range, IQR) for continuous variables, for which the normality of distribution was tested using the Kolmogorov–Smirnov test, and number (percentage, %) for categorical variables. Differences between groups for continuous variables and categorical variables were tested by using the analysis of variance and χ2 tests, respectively.

The prospective association of SES or healthy behaviours with outcomes was estimated using Cox proportional hazards regression and presented as hazard ratios (HRs) and 95% confidence intervals (CIs). We used Schoenfeld residuals to test the proportional hazards assumption, and calculated person years from baseline until the date of death from respiratory diseases, death by other causes, emigration, or the end of the follow-up period (30 June 2020), whichever occurred first. Model 1 included SES (high, medium, low), age (<65 years, ≥65 years), gender (male, female), race and ethnicity (White, Black, Asian, mixed, other), general health (excellent, good, fair, poor), weight loss (yes, no), diabetes (yes, no), cardiovascular disease (yes, no), cancer (yes, no), family history (yes, no, unknown), poor psychological status (yes, no), sleep duration (normal, short, long), coffee intake (yes, no), and consumption of tea (continuous), based on previous research [15]. Model 2 additionally included the healthy behaviours. A stratified analysis was conducted by healthy behaviours to investigate associations of the SES with health outcomes.

We quantified the additive and multiplicative interactions by adding a product term of SES (high, low) and healthy behaviours (no or one, four or five) in the model [15]. The interaction on the additive scale was measured by using the synergy index and corresponding 95% CI [24]. After classifying participants into nine groups according to SES (high, medium, low) and healthy behaviours score (no or one, two or three, four or five points), we estimated HRs of mortality in the different groups, compared with those of high SES and with four or five healthy behaviours, to assess the joint associations. To test the robustness and potential variations between subgroups, we repeated stratified analyses on gender and age groups.

In addition, we conducted two sensitivity analyses: (1) to test the influence of missing variables, we used multiple imputation by chained equations to impute all missing covariates; and (2) excluding participants who had an outcome event during the first 5 years of follow-up [15,25].

A two-sided *p* value of less than 0.05 was considered statistically significant. R software, version 4.2.1 was used for all analyses.

## 3. Results

### 3.1. Population Characteristics

Table 1 shows participants’ baseline characteristics. Among the 372,845 participants (mean age 56.03 years, 52.2% men), 78,104 (20.9%) were of high SES, 200,296 (53.7%) of medium SES, and 94,445 (25.3%) of low SES. Adults of low SES were more likely to be older. Men, non-White people, experiencing weight loss last year, a poorer general health, multiple comorbidities, abnormal sleep duration, less consumption of coffee, less healthy behaviours, abnormal BMI, and smoking were more prevalent among adults of low SES.

### 3.2. Associations of Healthy Behaviours and SES with Respiratory Disease Mortality

There were 1447 deaths from total respiratory diseases during the 12.47 median follow-up years (interquartile range [IQR], 11.56 to 13.28 years). The mortality was 0.01, 0.02, and 0.08 per 100 person-years among adults of high, medium, and low SES, respectively. After adjusting for healthy behaviours and other covariates, compared with high SES, low SES was associated with a 4.44-fold (95% CI: 3.42, 5.78) higher risk of total respiratory disease mortality. Adjusting for healthy behaviours had no effect on HR estimates (Table 2: 4.48 vs. 4.44). Results were not materially changed in sensitivity analyses (Appendix A). After adjusting for healthy behaviours and other covariates, compared with high SES, low SES was associated with a 2.64-fold and 7.20-fold higher risk of influenza and pneumonia mortality and chronic lower respiratory disease mortality, respectively (Appendix A shows that, among the various healthy behaviour subgroups, low SES was associated with higher risks of total respiratory disease mortality, whereas the associations were stronger in the no or one healthy behaviour subgroup. The results stratified by healthy behaviours for influenza, pneumonia, and chronic lower respiratory diseases were consistent with those for total respiratory diseases (Appendix A).

More healthy behaviours, compared with no or one healthy behaviour, were associated with 29% to 56% lower risk of total respiratory disease mortality, 39% to 41% lower risk of influenza and pneumonia mortality, and 29% to 45% lower risk of chronic lower respiratory disease mortality (Appendix A). Four or five healthy behaviours were associated with a lower risk of total respiratory disease mortality, whereas the associations were stronger in the low SES group (Appendix A). The results stratified by SES for influenza and pneumonia were consistent with those for total respiratory diseases, but for chronic lower respiratory diseases, the beneficial effect of more healthy behaviours was stronger among those of high SES (Appendix A).

### 3.3. Interaction and Joint Analysis of Healthy Behaviours and SES with Respiratory Disease Mortality

No significant multiplicative or additive interaction was found between healthy behaviours and SES on total respiratory disease mortality or influenza and pneumonia mortality, but there was an additive interaction between healthy behaviours and SES on chronic lower respiratory disease mortality (Table 2 and Appendix A). Compared with those of high SES and with four or five healthy behaviours, the HRs for individuals of low SES and with no or one healthy behaviour were 8.32 (95% CI: 4.23, 16.35), 6.68 (95% CI: 2.04, 21.87), and 29.90 (95% CI: 4.15, 215.35) for total respiratory disease mortality, influenza and pneumonia mortality, and chronic lower respiratory disease mortality, respectively (Figure 2 and Appendix A). Sensitivity analyses showed stable results (Appendix A).

### 3.4. Associations of Healthy Behaviours and SES with Respiratory Disease Mortality among Subpopulations

The associations of low SES with respiratory disease mortality were stronger in men than in women (Appendix A: 7.25 vs. 1.63) and in younger than older adults (Appendix A: 4.49 vs. 4.41). The results were similar for mortality from influenza and pneumonia and chronic lower respiratory diseases (Appendix A). Figure 3 showed no evidence of significant effect of the joint associations of healthy behaviours and SES on mortality from respiratory diseases for ages ≥ 65 years. In males and females, for groups including medium SES and no or one healthy behaviour, low SES and four or five healthy behaviours, low SES and two or three healthy behaviours, and low SES and no or one healthy behaviour, the estimated effects were statistically significant (in females, the groups additionally included medium SES and two or three healthy behaviours). The joint associations of healthy behaviours and SES with total respiratory disease mortality were stronger in men than in women and in younger than older adults (Figure 3). The results for mortality from influenza and pneumonia and chronic lower respiratory diseases are shown in Appendix A.

## 4. Discussion

To our knowledge, this was the first study to explore whether healthy behaviours affect the association of SES with respiratory disease mortality and examine the extent of joint relations between healthy behaviours and SES with respiratory disease mortality. We found that low SES was associated with a higher risk of mortality from respiratory disease, and the effect value was higher in the no or one healthy behaviour subgroup in this prospective analysis of nearly 400,000 participants from the UK Biobank. In addition, the joint analysis found that the highest risks of respiratory disease mortality were seen in adults of low SES and with no or one healthy behaviour, especially in adults aged ≤ 65 years and males.

Up to now, limited cohort studies have analysed the associations between SES and respiratory disease mortality. Our study explored the comprehensive associations between SES and respiratory disease mortality and found that adults of low SES had a higher risk of respiratory disease mortality. One cohort study from Poland reported that higher socioeconomic status was associated with a lower risk of mortality due to diseases of the respiratory system, which supports our results [8]. Our study acquired death information until 30 June 2020; the onset of the COVID-19 pandemic could have determined an increase in mortality from respiratory diseases starting from March–April 2020. It should be noted that the impact of the pandemic was stronger among the more vulnerable groups of the population (i.e., low-educated, low-income, and unhealthy behaviours) that were usually found at higher risk of infection and mortality [26,27]. Therefore, it is important to pay attention to the impact of SES inequalities on the progression of respiratory diseases and take measures to reduce the inequalities of SES or attenuate its harmful effects on respiratory disease mortality using relevant screening and intervention programmes. After adjusting for healthy behaviours, there was a minimal change in the HR for low SES, from 4.48 to 4.44. However, we observed joint associations between healthy behaviours and SES; at the same level of SES, the risk of respiratory disease mortality was lower among adults with more healthy behaviours. The risk of mortality from respiratory diseases was highest among persons of low SES who had no or one healthy behaviour. Previous studies reported that healthy behaviours, as important factors influencing health, might alleviate the risk of death [10,11,12]. Our results highlight that adherence to healthy behaviours represents potentially modifiable targets for improving the harmful impact of low SES on life expectancy from respiratory disease. The benefits of adopting healthy behaviors (two or more) are evident when considering the joint effect of SES levels and healthy behaviors.

Moreover, our study expanded on findings by showing age and gender differences for the associations of SES with mortality from respiratory diseases. We further identified that the associations were stronger in men than in women and in younger than older adults. Wang et al. also found that low SES had a larger effect on mortality among younger population [28]. Similarly, joint associations of less healthy behaviours and low SES with mortality from respiratory diseases were also stronger in men than in women, and in younger than older adults. The reasons for age and gender differences are not yet clear. It may be that men have higher respiratory disease mortality, and men and younger people with less healthy behaviours and/or of low SES may be more exposed to other risk factors for respiratory disease mortality, such as second-hand smoke [2,29]. Further research is needed to replicate this finding and identify the mechanisms behind the age and gender differences for the above associations. The above findings suggest that SES inequalities cause gaps in mortality from respiratory diseases, especially in younger male populations.

We also observed that experiencing weight loss in the past year and having multiple comorbidities, poorer general health, and abnormal sleep duration were more prevalent among adults of low SES. Other studies also found similar characters [15]. Adults of low SES with more comorbidities and poor health may explained the higher risk of respiratory diseases mortality in this group. Additionally, higher SES generally means more resources, including better living accommodations and medical care.

The large sample size and sufficient statistical power of our results were the major strengths of this study. Additionally, death records were obtained from the NHS Information Centre and the NHS Central Register to ensure accuracy. There were some limitations in the study. First, part of the self-reported information may cause recall bias. Thus, further objective measurements should be urgently carried out in this field. Second, although our estimation models adjusted for a wide range of known confounders, the potential for residual bias still existed. Third, the healthy behaviour index was defined by the assessment of five factors: alcohol intake, smoking status, physical exercise, eating habits, and BMI in UKB studies. Although BMI is not a behaviour and is commonly used as a marker of healthy behaviour (low level of physical activity, poor diet) in other studies, it may represent other potential health behaviours and help to reflect the subjects’ healthy behaviours comprehensively. In addition, although SES was evaluated by three factors, including employment status, income, and educational attainment, using latent class analysis in this study, it still cannot represent SES comprehensively. For example, the employed group included those in paid employment or self-employment, retired, doing unpaid work, voluntary workers, and student categories, but doing unpaid work, voluntary workers, and student categories might have different health outcomes compared to income perceivers. Finally, results cannot be generalised because of the convenience of the sample, regardless of its size and statistical power.

## 5. Conclusions

In conclusion, this large nationwide UK prospective cohort analysis found that both low SES and less healthy behaviours were associated with an increased risk of respiratory disease mortality, which augmented when both presented together. Our results highlight the risk of SES inequalities and unhealthy behaviours for the progression of respiratory disease, especially their joint effect. In the future, understanding their mutual relationship in reducing respiratory disease burden will still be required by using other approaches, such as a randomised controlled trial. Our novel findings may provide relevant evidence for respiratory disease management in clinical and public health.

## Figures and Tables

**Figure 1 nutrients-15-01872-f001:**
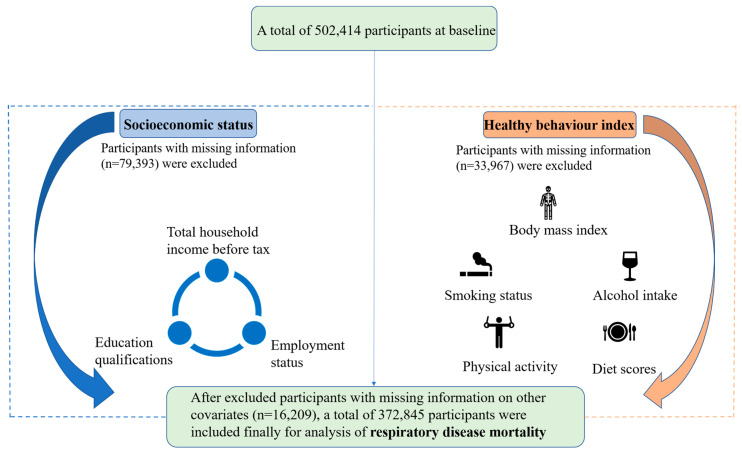
The study flowchart.

**Figure 2 nutrients-15-01872-f002:**
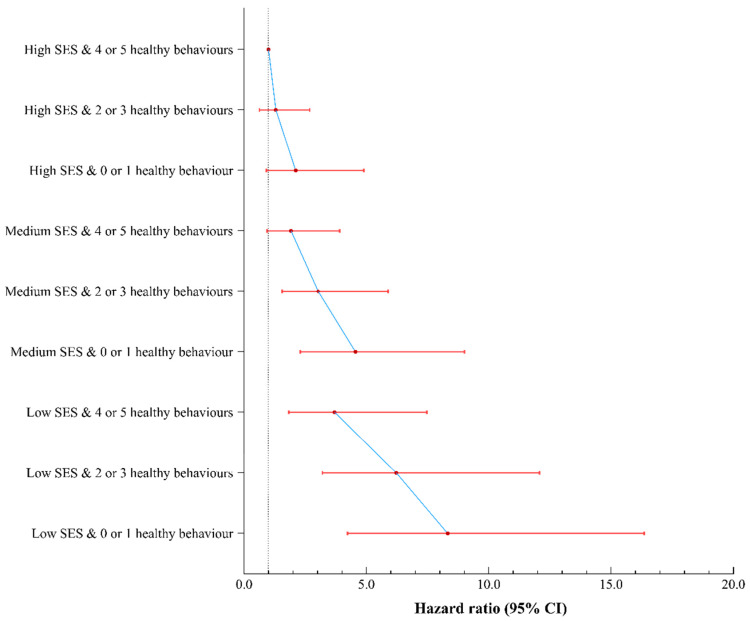
Joint associations of healthy behaviors and socioeconomic status with total respiratory disease mortality. Models all adjusted for age, gender, race and ethnicity, general health, weight loss, diabetes, cardiovascular disease, cancer, family history, poor psychological status, sleep duration, coffee intake, and consumption of tea. 95% CI = 95% confidence interval. SES = socioeconomic status.

**Figure 3 nutrients-15-01872-f003:**
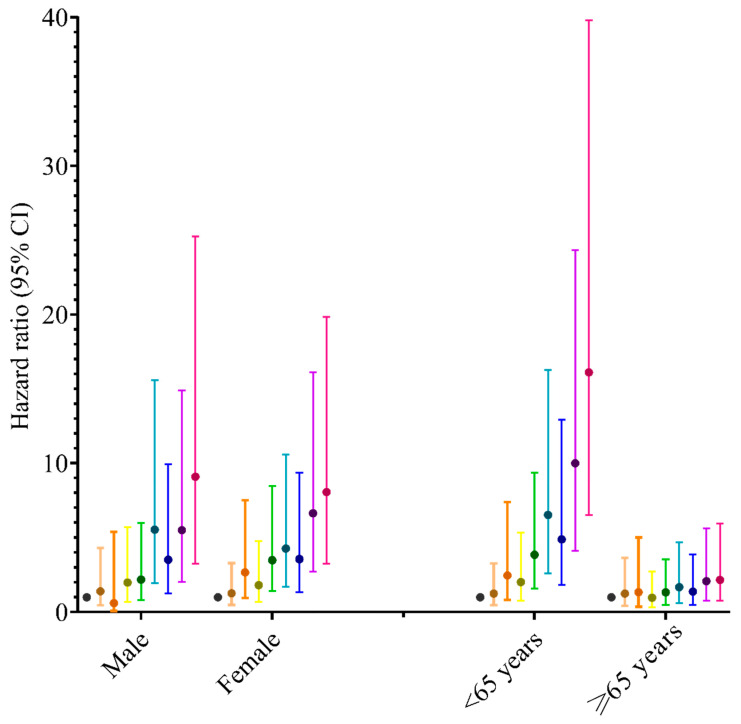
The joint associations of healthy behaviours and socioeconomic status on respiratory disease mortality by gender and age. Models all adjusted for age, gender, race and ethnicity, general health, weight loss, diabetes, cardiovascular disease, cancer, family history, poor psychological status, sleep duration, coffee intake, and consumption of tea. Nine joint groups are presented in the following order: High SES and four or five healthy behaviours, high SES and two or three healthy behaviours, high SES and no or one healthy behaviour, medium SES and four or five healthy behaviours, medium SES and two or three healthy behaviours, medium SES and no or one healthy behaviour, low SES and four or five healthy behaviours, low SES and two or three healthy behaviours, low SES and no or one healthy behaviour. 95% CI = 95% confidence interval. SES = socioeconomic status.

**Table 1 nutrients-15-01872-t001:** Baseline characteristics of participants.

Characteristics	N	High SES (n, %)	Medium SES (n, %)	Low SES (n, %)
	372,845	78,104 (20.9)	200,296 (53.7)	94,445 (25.3)
Mean age (SD), y	56.03 (8.07)	52.48 (7.20)	55.76 (7.98)	59.52 (7.49)
Age, y				
<65	309,375 (83.0)	74,020 (94.8)	169,570 (84.7)	65,785 (69.7)
≥65	63,470 (17.0)	4084 (5.2)	30,726 (15.3)	28,660 (30.3)
Gender				
Male	194,555 (52.2)	37,761 (48.3)	103,574 (51.7)	53,220 (56.4)
Female	178,290 (47.8)	40,343 (51.7)	96,722 (48.3)	41,225 (43.6)
Race and ethnicity				
White	357,423 (95.9)	75,115 (96.2)	192,582 (96.1)	89,726 (95.0)
Black	4537 (1.2)	495 (0.6)	2500 (1.2)	1542 (1.6)
Asian	6288 (1.7)	1447 (1.9)	3036 (1.5)	1805 (1.9)
Mixed	2052 (0.6)	514 (0.7)	1000 (0.5)	538 (0.6)
other	2545 (0.7)	533 (0.7)	1178 (0.6)	834 (0.9)
General health				
Excellent	67,336 (18.1)	21,086 (27.0)	35,577 (17.8)	10,673 (11.3)
Good	219,590 (58.9)	46,077 (59.0)	123,149 (61.5)	50,364 (53.3)
Fair	72,234 (19.4)	9803 (12.6)	36,775 (18.4)	25,656 (27.2)
Poor	13,685 (3.7)	1138 (1.5)	4795 (2.4)	7752 (8.2)
Weight loss				
No	315,831 (84.7)	66,706 (85.4)	170,194 (85.0)	78,931 (83.6)
Yes	57,014 (15.3)	11,398 (14.6)	30,102 (15.0)	15,514 (16.4)
Cancer				
No	344,912 (92.5)	73,520 (94.1)	185,777 (92.8)	85,615 (90.7)
Yes	27,933 (7.5)	4584 (5.9)	14,519 (7.2)	8830 (9.3)
Diabetes				
No	355,309 (95.3)	76,011 (97.3)	192,006 (95.9)	87,292 (92.4)
Yes	17,536 (4.7)	2093 (2.7)	8290 (4.1)	7153 (7.6)
Poor psychological status				
No	244,873 (65.7)	56,290 (72.1)	133,101 (66.5)	55,482 (58.7)
Yes	127,972 (34.3)	21,814 (27.9)	67,195 (33.5)	38,963 (41.3)
Cardiovascular disease				
No	267,234 (71.7)	63,133 (80.8)	146,167 (73.0)	57,934 (61.3)
Yes	105,611 (28.3)	14,971 (19.2)	54,129 (27.0)	36,511 (38.7)
Family history				
No	29,663 (8.0)	7739 (9.9)	15,636 (7.8)	6288 (6.7)
Yes	335,046 (89.9)	69,224 (88.6)	180,654 (90.2)	85,168 (90.2)
Unknown	8136 (2.2)	1141 (1.5)	4006 (2.0)	2989 (3.2)
Sleep duration				
Normal	273,210 (73.3)	60,963 (78.1)	149,144 (74.5)	63,103 (66.8)
Short	88,334 (23.7)	16,505 (21.1)	46,557 (23.2)	25,272 (26.8)
Long	11,301 (3.0)	636 (0.8)	4595 (2.3)	6070 (6.4)
Tea intake (median [IQR]), cups/day	3.00 [1.00, 5.00]	3.00 [1.00, 5.00]	3.00 [1.00, 5.00]	3.00 [2.00, 5.00]
Coffee intake				
No	79,050 (21.2)	13,736 (17.6)	41,457 (20.7)	23,857 (25.3)
Yes	293,795 (78.8)	64,368 (82.4)	158,839 (79.3)	70,588 (74.7)
Healthy behaviours				
No or one healthy behaviour	40,478 (10.9)	8035 (10.3)	21,348 (10.7)	11,095 (11.7)
Two or three healthy behaviours	252,202 (67.6)	50,053 (64.1)	135,701 (67.8)	66,448 (70.4)
Four or five healthy behaviours	80,165 (21.5)	20,016 (25.6)	43,247 (21.6)	16,902 (17.9)
Body mass index (kg/m^2^)				
<18.5/>24.9	252,073 (67.6)	47,520 (60.8)	136,258 (68.0)	68,295 (72.3)
18.5–24.9	120,772 (32.4)	30,584 (39.2)	64,038 (32.0)	26,150 (27.7)
Smoking				
Yes	168,194 (45.1)	29,500 (37.8)	88,892 (44.4)	49,802 (52.7)
No	204,651 (54.9)	48,604 (62.2)	111,404 (55.6)	44,643 (47.3)
Diet score				
Zero–three	315,270 (84.6)	66,428 (85.1)	169,098 (84.4)	79,744 (84.4)
Four–seven	57,575 (15.4)	11,676 (14.9)	31,198 (15.6)	14,701 (15.6)
Physical activity				
Insufficient	73,412 (19.7)	17,477 (22.4)	38,608 (19.3)	17,327 (18.3)
Sufficient	299,433 (80.3)	60,627 (77.6)	161,688 (80.7)	77,118 (81.7)
Alcohol intake				
Inappropriate	44,392 (11.9)	10,481 (13.4)	24,028 (12.0)	9883 (10.5)
Moderate	328,453 (88.1)	67,623 (86.6)	176,268 (88.0)	84,562 (89.5)

Notes: all *p*-values < 0.0001; SES = socioeconomic status.

**Table 2 nutrients-15-01872-t002:** Associations of socioeconomic status with total respiratory disease mortality.

	Deaths/Mortality (per 100 Person-Years)	Hazard Ratio (95% CI) *	
		Unadjusted for Healthy Behaviours	Adjusted for Healthy Behaviours ^†^
High SES	64/0.01	1 (Reference)	1 (Reference)
Medium SES	522/0.02	2.23 (1.72, 2.90)	2.23 (1.72, 2.90)
Low SES	861/0.08	4.48 (3.45, 5.82)	4.44 (3.42, 5.78)

Models all adjusted for age, gender, race and ethnicity, general health, weight loss, diabetes, cardiovascular disease, cancer, family history, poor psychological status, sleep duration, coffee intake, and consumption of tea. * Hazard ratios for the product term between the healthy behaviours (no or one v four or five) and SES (high v low) were used to evaluate multiplicative interaction, and its confidence interval, which did not include 1, meant the statistically significant multiplicative interaction. The synergy index between the healthy behaviours (no or one vs. four or five) and SES (high vs. low) was used to evaluate additive interaction, and its confidence interval did not include 1 meant the statistically significant additive interaction [15,24]. ^†^ Multiplicative interaction: 1.06 (95% CI: 0.44, 2.59), *p* = 0.892; additive interaction: the synergy index = 1.16 (95% CI: 0.69, 1.97). SES = socioeconomic status.

## Data Availability

The UK Biobank datasets are openly available by submitting a data request proposal from https://www.ukbiobank.ac.uk/ (accessed on 9 June 2022). We are authorised to access the database through the Access Management System (AMS) (Application number: 79114).

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
