# Peer review of "Mutual Associations of Healthy Behaviours and Socioeconomic Status with Respiratory Diseases Mortality: A Large Prospective Cohort Study"

_nutrients, 2023, doi:10.3390/nu15081872_

Round 1

Reviewer 1 Report

This study is a prospective cohort including 372845 participants to investigate the mutual association of healthy behavior on socio-economic status (SES) on respiratory disease mortality.

The study finds that low SES was associated with a higher risk of respiratory disease mortality.  The risk of respiratory disease mortality was higher in adult of low SES with no healthy behavior. The authors report no significant multiplicative and additive interaction between healthy behaviors and SES on respiratory disease mortality. Overall, the large sample size and results in this study are interesting and highlight than healthy behavior can help reduce the gap in respiratory disease mortality associated with economic status. 

There are however points that would benefit to be addressed by the authors prior to publication, notably expanding on potential limitation of the study.

Healthy behavior index was defined by the assessment of 5 factors: alcohol intake, smoking status, physical exercise, eating habits and BMI. Limitations of this strategy should be mentioned.

While body mass index may be indicative of health, it is not necessarily a behavior. A high body mass index may be a result of the other categories that are used as markers of healthy behavior (low level of physical activity, poor diet) and author should discuss whether this could be a limitation.

Additionally, when looking at respiratory disease mortality, some of the healthy behaviors may be more critical than other in their association with disease mortality (smoking is particularly associated with chronic respiratory diseases such as COPD): have the authors looked whether some healthy/ unhealthy behaviors were more common in the different SES groups?

Both acute and chronic respiratory illnesses are included in the International Statistical Classification of Diseases and 129 Related Health Problems, 10th Revision (ICD-10): J00-J99 definition. Was respiratory disease mortality mainly linked to chronic respiratory conditions?  If smoking status is different in the SES group, could the authors comment of the type of respiratory disease associated which each SES, in the even where some of the healthy behaviors were different in each SES groups? 

Finally, minor spell check is required before publication (few examples below):

line 25 : HR abbreviation should be defined 

line 177 : 'morality' should be ' mortality'

line 208 : " healthy' is repeated twice

line 243 : we found that low SES could increase the 4.48 times higher risk of respiratory diseases mortality.

Author Response

Reviewer(s)' Comments to Author:

Reviewer: 1

This study is a prospective cohort including 372845 participants to investigate the mutual association of healthy behavior on socio-economic status (SES) on respiratory disease mortality.

The study finds that low SES was associated with a higher risk of respiratory disease mortality.  The risk of respiratory disease mortality was higher in adult of low SES with no healthy behavior. The authors report no significant multiplicative and additive interaction between healthy behaviors and SES on respiratory disease mortality. Overall, the large sample size and results in this study are interesting and highlight than healthy behavior can help reduce the gap in respiratory disease mortality associated with economic status. There are however points that would benefit to be addressed by the authors prior to publication, notably expanding on potential limitation of the study.

Response: Thanks.

Healthy behavior index was defined by the assessment of 5 factors: alcohol intake, smoking status, physical exercise, eating habits and BMI. Limitations of this strategy should be mentioned. While body mass index may be indicative of health, it is not necessarily a behavior. A high body mass index may be a result of the other categories that are used as markers of healthy behavior (low level of physical activity, poor diet) and author should discuss whether this could be a limitation.

Response: Thanks for your good suggestion. Although BMI is not a behavior and are used as markers of healthy behavior (low level of physical activity, poor diet), it may represent other potential health behaviors. We used the healthy behavior index based on previous UKB studies which used this index (Yang G, Cao X, Li X, Zhang J, Ma C, Zhang N, Lu Q, Crimmins EM, Gill TM, Chen X et al: Association of Unhealthy Lifestyle and Childhood Adversity With Acceleration of Aging Among UK Biobank Participants. JAMA network open 2022, 5(9):e2230690; Bountziouka V, Musicha C, Allara E, Kaptoge S, Wang Q, Angelantonio ED, Butterworth AS, Thompson JR, Danesh JN, Wood AM et al: Modifiable traits, healthy behaviours, and leukocyte telomere length: a population-based study in UK Biobank. The Lancet Healthy longevity 2022, 3(5):e321-e331). We have indicted this limitation as shown in page 10 line 324-328: “Thirdly, healthy behavior index was defined by the assessment of five factors: alcohol intake, smoking status, physical exercise, eating habits and BMI in UKB studies. Alt-hough BMI is not a behavior and commonly used as markers of healthy behavior (low level of physical activity, poor diet) in other studies, it may represent other potential health behaviors and help to reflect the subjects’ healthy behavior comprehensively.”.

Additionally, when looking at respiratory disease mortality, some of the healthy behaviors may be more critical than other in their association with disease mortality (smoking is particularly associated with chronic respiratory diseases such as COPD): have the authors looked whether some healthy/ unhealthy behaviors were more common in the different SES groups?

Response: Thanks for your good suggestion, we have added the distribution of healthy behaviors in the different SES groups. We found that less healthy behaviours, abnormal BMI and smoking were more prevalent among adults of low SES, we have added it in results as shown in page 5 line 178-181 and table 1: “Men, non-white people, experiencing weight loss last year, a poorer general health, multiple comorbidities, abnormal sleep duration, less consumption of coffee, less healthy behaviours, abnormal BMI and smoking were more prevalent among adults of low SES”.

Both acute and chronic respiratory illnesses are included in the International Statistical Classification of Diseases and 129 Related Health Problems, 10th Revision (ICD-10): J00-J99 definition. Was respiratory disease mortality mainly linked to chronic respiratory conditions? If smoking status is different in the SES group, could the authors comment of the type of respiratory disease associated which each SES, in the even where some of the healthy behaviors were different in each SES groups?

Response: Thanks for your good suggestion. We have supplemented related analysis on influenza and pneumonia and chronic lower respiratory diseases. After adjusting for healthy behaviours and other covariates, compared with adults of high SES, adults of low SES were associated with 2.64 times and 7.20 times higher risk of mortality from influenza and pneumonia and chronic lower respiratory diseases, respectively. The results stratified by healthy behaviours for influenza and pneumonia and chronic lower respiratory diseases were consistent with total respiratory diseases. More healthy behaviours, compared with no or one healthy behaviour, were associated with 29% to 56% lower risks of total respiratory diseases mortality; 39% to 41% lower risks of influenza and pneumonia mortality; 29% to 45% lower risks of chronic lower respiratory diseases mortality. The results stratified by SES for influenza and pneumonia was consistent with total respiratory diseases, but for chronic lower respiratory diseases, the associations were stronger among those of high SES. Hazard ratios for individuals of low SES and with no or one healthy behaviours, compared with those of high SES and with four or five healthy behaviours were 8.32 (95% CI: 4.23, 16.35), 6.68 (95% CI: 2.04, 21.87) and 29.90 (95% CI: 4.15, 215.35) for total respiratory diseases mortality, influenza and pneumonia mortality; and chronic lower respiratory diseases mortality, respectively.  We have added related information in results and revised methods.

Finally, minor spell check is required before publication (few examples below):

line 25 : HR abbreviation should be defined

Response: Thanks. We have defined HR abbreviation.

line 177 : 'morality' should be ' mortality'

Response: Thanks. We have corrected it and reviewed this full article.

line 208 : " healthy' is repeated twice

Response: Thanks. We have deleted it and reviewed this full article.

line 243 : we found that low SES could increase the 4.48 times higher risk of respiratory diseases mortality.

Response: Thanks. We have restated this sentence.

Reviewer 2 Report

Dear author,

I just read an excellent manuscript aiming to investigate whether healthy behaviours affect association of SES with respiratory diseases mortality and examine the extent of joint relations of healthy behaviours and SES with respiratory diseases mortality.

I have a few commnets to make:

Regarding the study population, was ther any stratification in sampling according to rural/ urban areas, as well as to the whole population of each area in UK from where the sample was taken?

In the paragraph Assessment of SES, all abbreviations must be explained, eg. General Certificate of Secondary Education (GCSE).

In Lines 128-130 the proper reference must be added regarding ICD-10.

In Line 132, Mean (SD) was used for all continuous variables, regardless the normality of distribution? Or all continous variables had normal distribution. Please mention the test you used for defining that.

The statistical analysis and the presentation of your results is excellent. The used plots help the reader to grasp the extent of the investigated associations.

Although the manuscript is well written, the Discussion has many grammatical errors. Please revise it accordingly.

In the Strengths and Limitations paragraph, I believe that it should be added that the results cannnot be generalized because of the convenient used sample, regardless its size and statistical power.

Author Response

Reviewer: 2

Dear author,

I just read an excellent manuscript aiming to investigate whether healthy behaviours affect association of SES with respiratory diseases mortality and examine the extent of joint relations of healthy behaviours and SES with respiratory diseases mortality.

Response: Thanks.

I have a few commnets to make:

Regarding the study population, was ther any stratification in sampling according to rural/ urban areas, as well as to the whole population of each area in UK from where the sample was taken?

Response: Thanks, the participants were assessed in 22 assessment centres in Scotland, England and Wales throughout the UK, covering a variety of different settings to provide socioeconomic and ethnic heterogeneity and urban–rural mix. (https://www.ukbiobank.ac.uk/enable-your-research/about-our-data/baseline-assessment; Sudlow C, Gallacher J, Allen N, Beral V, Burton P, Danesh J, Downey P, Elliott P, Green J, Landray M, Liu B, Matthews P, Ong G, Pell J, Silman A, Young A, Sprosen T, Peakman T, Collins R. UK biobank: an open access resource for identifying the causes of a wide range of complex diseases of middle and old age. PLoS Med. 2015 Mar 31;12(3):e1001779. doi: 10.1371/journal.pmed.1001779).

In the paragraph Assessment of SES, all abbreviations must be explained, eg. General Certificate of Secondary Education (GCSE).

Response: Thanks, we have added all abbreviations in the paragraph Assessment of SES as shown in page 3 line 94-100: “Education qualifications included eight options: college or university degree; advanced (A) levels , advanced subsidiary (AS) levels, or equivalent; general certification of ed-ucation ordinary (O) level, the general certificate of secondary education (GCSEs), or equivalent; the certificate of secondary education (CSEs) or equivalent; national voca-tional qualification (NVQ), higher national diploma (HND), higher national certificate (HNC), or equivalent; other professional qualifications; none of the above (equivalent to less than high school diploma); or prefer not to answer.”.

In Lines 128-130 the proper reference must be added regarding ICD-10.

Response: Thanks, we have added the proper reference “International Statistical Classification of Diseases and Related Health Problems 10th Revision. 2019. https://icd.who.int/browse10/2019/en”.

In Line 132, Mean (SD) was used for all continuous variables, regardless the normality of distribution? Or all continous variables had normal distribution. Please mention the test you used for defining that.

Response: Thanks, we have tested the continuous variables including age and tea consumption using Kolmogorov-Smirnov test. We have added it in methods as shown in page 4 line 138-140: “Baseline characteristics were presented as mean (standard deviation, SD) or me-dian (interquartile range, IQR) for continuous variables which of the normality of dis-tribution was tested using Kolmogorov-Smirnov test, and number (percentage, %) for categorical variables.”.

The statistical analysis and the presentation of your results is excellent. The used plots help the reader to grasp the extent of the investigated associations.

Response: Thanks.

Although the manuscript is well written, the Discussion has many grammatical errors. Please revise it accordingly.

Response: Thanks. We have revised the full article.

In the Strengths and Limitations paragraph, I believe that it should be added that the results cannnot be generalized because of the convenient used sample, regardless its size and statistical power.

Response: Thanks. We have added it in limitation as shown in page 10 line 334-336: “Finally, results cannot be generalized because of the convenient used sample, regardless its sample size and statistical power.”.

Reviewer 3 Report

I thank the editor for the opportunity to review this article. It is a large cohort with robust data.

Just a few comments:

Major comments

1. Although the authors mention in the discussion the differences between sex and age, they do not sufficiently explore the possible explanations that could exist. I suggest expanding the possible causes.

2. In addition, the respiratory causes of death vary between the general population that suffers an acute event (such as pneumonia) or those that already have chronic respiratory diseases (such as COPD or Asthma) and who could have died from respiratory causes. Was it analyzed according to chronic respiratory comorbidity?

Minor comments

1.       Authors may abbreviate chronic respiratory diseases as CRD

2. In line 208, the word healthy is repeated

3.       In figure 2, use the same format for abbreviations. The authors use : and = to explain them

Author Response

Reviewer 3:

I thank the editor for the opportunity to review this article. It is a large cohort with robust data.

Response: Thanks.

Just a few comments:

Major comments

  1. Although the authors mention in the discussion the differences between sex and age, they do not sufficiently explore the possible explanations that could exist. I suggest expanding the possible causes.

Response: Thanks. We have expanded the possible causes for the differences between sex and age in the discussion as shown in page 9 line 299-311: “Moreover, our study expanded on findings by showing age and sex differences for the associations of SES with mortality from respiratory diseases. We further identified that the association were stronger in men than in women, and in younger than older adults. Wang et al. also found that the low SES had larger effect on mortality among younger population [28]. Similarly, joint associations of less healthy behaviours and low SES on mortality from respiratory diseases were also stronger in men than in women, and in younger than older adults. The reasons for age and sex differences are not yet clear. It may be that men have higher respiratory disease mortality, and men and younger people with less healthy behaviours or/and in low SES may be exposed to other risk factors for respiratory disease mortality, such as second-hand smoke easier and more [2,29]. Further research is needed to replicate this finding and identify the mechanisms behind the age and sex differences for above associations. Above findings suggest that SES inequalities cause gap of mortality from respiratory diseases, espe-cially in younger male populations.”.

  1. In addition, the respiratory causes of death vary between the general population that suffers an acute event (such as pneumonia) or those that already have chronic respiratory diseases (such as COPD or Asthma) and who could have died from respiratory causes. Was it analyzed according to chronic respiratory comorbidity?

Response: Thanks for your good suggestion. We have supplemented related analysis on influenza and pneumonia and chronic lower respiratory diseases. After adjusting for healthy behaviours and other covariates, compared with adults of high SES, adults of low SES were associated with 2.64 times and 7.20 times higher risk of mortality from influenza and pneumonia and chronic lower respiratory diseases, respectively. The results stratified by healthy behaviours for influenza and pneumonia and chronic lower respiratory diseases were consistent with total respiratory diseases. More healthy behaviours, compared with no or one healthy behaviour, were associated with 29% to 56% lower risks of total respiratory diseases mortality; 39% to 41% lower risks of influenza and pneumonia mortality; 29% to 45% lower risks of chronic lower respiratory diseases mortality. The results stratified by SES for influenza and pneumonia was consistent with total respiratory diseases, but for chronic lower respiratory diseases, the associations were stronger among those of high SES. Hazard ratios for individuals of low SES and with no or one healthy behaviours, compared with those of high SES and with four or five healthy behaviours were 8.32 (95% CI: 4.23, 16.35), 6.68 (95% CI: 2.04, 21.87) and 29.90 (95% CI: 4.15, 215.35) for total respiratory diseases mortality, influenza and pneumonia mortality; and chronic lower respiratory diseases mortality, respectively.  We have added related information in results and revised methods.

Minor comments

  1. Authors may abbreviate chronic respiratory diseases as CRD

Response: Thanks. We have abbreviated chronic respiratory diseases as CRD and review this full article.

  1. In line 208, the word healthy is repeated

Response: Thanks. We have deleted it and reviewed this full article.

  1. In figure 2, use the same format for abbreviations. The authors use : and = to explain them

Response: Thanks. We have used = to explain them in figure 2 and 3.

Reviewer 4 Report

The study investigated mutual associations of healthy behaviours and socioeconomic status with respiratory diseases mortality using cohort data with large sample size from the UK Biobank. The paper is relevant for the analysis of the impact of healthy behavior on socioeconomic inequalities in mortality from respiratory diseases. The results are interesting, but some findings should be revised. The methodology used is appropriate, however some clarifications are required and some aspects should be improved.

My comments and suggestions to authors:

·         Introduction, p.2. The last paragraph (lines 65-70) of the introduction is a bit confusing. I suggest to rephrase the last two sentences to allow a more immediate comprehension of study goals by the reader. In particular the aims of the study should be more clearly declared and the sentence reorganized (it might be useful distinguishing between main and secondary goals) for instance starting like this: primary goal is to investigate the impact of the association among SES and healthy behaviours on mortality from respiratory diseases by sex and age-group)….

·         Table S1. Number of participants with missing covariates: Please check the numbers and percentages reported in the table. For healthy behaviours n=52910 while both in the text and Figure 1 the authors reported n=33967. In addition, please specify in the heading of the table to which total the percentage refers to. Is the percentage calculated on the total number of the individuals of the cohort? I don’t think so as for Socioeconomic status (n=79393) we should have 15.8%.

·         Figure 1. The left side pertain to Socioeconomic status and the right side to Healthy behaviors. In my opinion this information should be reported at the top of the figure, in each side. It is unclear what function do the expressions “low socioeconomic status” and “more healthy behaviours” have in the flowchart. I suggest to explain what the two expressions refer to or to remove them from the Figure.

·         Methods, p.3 Lines 92-95: categories of Education qualification are listed using abbreviations. Many readers will not be familiar with such categories, thus provide complete description of each category (reporting abbreviations in brackets)

·         Appendix 1: The inclusion among the employed group of “doing unpaid”, “voluntary workers” and students categories is questionable, as these individuals might have different health outcomes compared to income perceivers. At least the authors should address this point in the discussion. Looking at the table of Item-response probabilities (appendix 1) the variable “employment status” appears to be not determinant for identifying latent classes for SES and the use of income and educational attainment only will probably lead to similar results, through a more solid approach. My suggestion is to perform the analysis considering this approach.

·         Methods, p.4, lines 127-128: the commonly used term in official statistics is underlying cause of death (remove the term “primary”)

·         Methods, Statistical analysis p. 4 lines 139-140: “Person years were calculated from baseline until the date of death from respiratory diseases, or end of follow-up, whichever occurred first.” End of follow-up should include: death for other causes, emigration, end of follow-up period (30 june 2020?). This needs to be clarified and explicitly reported in the text.

·         Table 1. P-values from chi-square test for differences group indicates that differences among groups are always significant. This is an obvious consequence of the frequencies in the groups compared. I suggest to remove the column reporting p-values and to report in a table footnote that p-values are always  <0.0001

·         Results, p. 6, line 181-182: “the hazard ratios without adjustment for healthy behaviours are larger…”. It would be better to say that adjusting for the healthy behavior had no effect on HRs estimates (from Table 2: 4.48 vs 4.44; same also for estimates reported in Table S6)

·         Figure S1: please specify the reference categories for HRs  in the table footnote

·         Results, lines 217-222 - Figure3 showed no evidence of significant effect of the joint associations of healthy behaviours and SES on mortality from respiratory diseases for ages ≥65 years. Both in males and females it seems that only for groups from 6 to 9 the estimated effects are statistically significant. These evidences should be highlighted in the text.

·         Discussion,  p.8 line 238-239 “ highest risks of respiratory diseases mortality were seen in adults of low SES and no or one healthy behaviours” the finding refers to adults aged ≤65 years

·         Discussion, p.8-9 lines 245-248 the authors emphasized the role of healthy behavior, but this is not evident from the result they are mentioning, i.e. a change in the HR for low SES from 4.48 to 4.44 after adjusting for healthy behaviors. From this specific analysis the impact of healthy behaviors on mortality outcomes according to SES level is minimal. I would say that there is no impact. This part of the text should be revised according to this consideration. The benefits of adopting healthy behaviors (two or more) are evident from the analysis of joint associations with SES levels, as pointed out by the authors in the following part of the discussion.

·         The onset of the COVID-19 pandemic could have determined an increase in mortality from respiratory diseases staring from march-april 2020. It is known that the impact of the pandemic was stronger among the more vulnerable groups of the population (i.e. low educated) that were usually found at higher risk of infection and mortality. Even if the pandemic affects only a small time frame of the study period and likely do not affect the estimates, I think this aspect should be briefly addressed in the discussion.

·         The paper need moderate english language revision

Author Response

Review 4:

The study investigated mutual associations of healthy behaviours and socioeconomic status with respiratory diseases mortality using cohort data with large sample size from the UK Biobank. The paper is relevant for the analysis of the impact of healthy behavior on socioeconomic inequalities in mortality from respiratory diseases. The results are interesting, but some findings should be revised. The methodology used is appropriate, however some clarifications are required and some aspects should be improved.

Response: Thanks.

My comments and suggestions to authors:

  • Introduction, p.2. The last paragraph (lines 65-70) of the introduction is a bit confusing. I suggest to rephrase the last two sentences to allow a more immediate comprehension of study goals by the reader. In particular the aims of the study should be more clearly declared and the sentence reorganized (it might be useful distinguishing between main and secondary goals) for instance starting like this: primary goal is to investigate the impact of the association among SES and healthy behaviours on mortality from respiratory diseases by sex and age-group)

Response: Thanks for your good suggestions. We have rephrased the last two sentences as shown in page 2 line 67-71: “In this study, we aim to accomplish two goals by using cohort data with large sample size from the UK Biobank. The primary goal is to investigate the impact of the association among SES and healthy behaviours on mortality from respiratory diseases, and second goal is to explore whether findings are consistent among subpopulations by sex and age-group”.

  • Table S1. Number of participants with missing covariates: Please check the numbers and percentages reported in the table. For healthy behaviours n=52910 while both in the text and Figure 1 the authors reported n=33967. In addition, please specify in the heading of the table to which total the percentage refers to. Is the percentage calculated on the total number of the individuals of the cohort? I don’t think so as for Socioeconomic status (n=79393) we should have 15.8%.

Response: Thanks. We have corrected tableS1 healthy behaviours n=33,967, corrected the percentage as 15.8%; and specify in the heading of the table to which total the percentage refers to as shown in table S1.

  • Figure 1. The left side pertain to Socioeconomic status and the right side to Healthy behaviors. In my opinion this information should be reported at the top of the figure, in each side. It is unclear what function do the expressions “low socioeconomic status” and “more healthy behaviours” have in the flowchart. I suggest to explain what the two expressions refer to or to remove them from the Figure.

Response: Thanks for your good suggestions. We have reported “Socioeconomic status” and “Healthy behaviors” at the top of the figure in each side; and removed low socioeconomic status” and “more healthy behaviours”.

  • Methods, p.3 Lines 92-95: categories of Education qualification are listed using abbreviations. Many readers will not be familiar with such categories, thus provide complete description of each category (reporting abbreviations in brackets)

Response: Thanks, we have added all abbreviations in the paragraph Assessment of SES as shown in page 3 line 94-100: “Education qualifications included eight options: college or university degree; advanced (A) levels , advanced subsidiary (AS) levels, or equivalent; general certification of ed-ucation ordinary (O) level, the general certificate of secondary education (GCSEs), or equivalent; the certificate of secondary education (CSEs) or equivalent; national voca-tional qualification (NVQ), higher national diploma (HND), higher national certificate (HNC), or equivalent; other professional qualifications; none of the above (equivalent to less than high school diploma); or prefer not to answer.”.

  • Appendix 1: The inclusion among the employed group of “doing unpaid”, “voluntary workers” and students categories is questionable, as these individuals might have different health outcomes compared to income perceivers. At least the authors should address this point in the discussion. Looking at the table of Item-response probabilities (appendix 1) the variable “employment status” appears to be not determinant for identifying latent classes for SES and the use of income and educational attainment only will probably lead to similar results, through a more solid approach. My suggestion is to perform the analysis considering this approach.

Response: Thanks, although “employment status, income and educational attainment” cannot represent SES comprehensively, we used latent class analysis to classified SES based on the previous studies which using UKB data (Zhang YB, Chen C, Pan XF, Guo J, Li Y, Franco OH, Liu G, Pan A. Associations of healthy lifestyle and socioeconomic status with mortality and incident cardiovascular disease: two prospective cohort studies. BMJ. 2021 Apr 14;373:n604. doi: 10.1136/bmj.n604), we have added it in limitation as shown in page 10 line 329-334: “In addition, although SES was evaluated by three factors including employment status, income and educational attainment using latent class analysis in this study, it still cannot represent SES comprehensively. For example, the employed group included those in paid employment or self-employed, retired, doing unpaid, voluntary workers and students categories, but doing unpaid, voluntary workers and students categories might have different health outcomes compared to income perceivers.”.

  • Methods, p.4, lines 127-128: the commonly used term in official statistics is underlying cause of death (remove the term “primary”)

Response: Thanks, we have removed the term “primary”.

  • Methods, Statistical analysis p. 4 lines 139-140: “Person years were calculated from baseline until the date of death from respiratory diseases, or end of follow-up, whichever occurred first.” End of follow-up should include: death for other causes, emigration, end of follow-up period (30 june 2020?). This needs to be clarified and explicitly reported in the text.

Response: Thanks, we have clarified it in the methods.

  • Table 1. P-values from chi-square test for differences group indicates that differences among groups are always significant. This is an obvious consequence of the frequencies in the groups compared. I suggest to remove the column reporting p-values and to report in a table footnote that p-values are always <0.0001

Response: Thanks, we have removed the column reporting p-values and to report in a table footnote that p-values are always <0.0001.

  • Results, p. 6, line 181-182: “the hazard ratios without adjustment for healthy behaviours are larger…”. It would be better to say that adjusting for the healthy behavior had no effect on HRs estimates (from Table 2: 4.48 vs 4.44; same also for estimates reported in Table S6)

Response: Thanks, we have changed this sentence as “Adjusting for the healthy behavior had no effect on HRs estimates (Table 2: 4.48 vs 4.44).”

  • Figure S1: please specify the reference categories for HRs in the table footnote

Response: Thanks, we have specified the reference categories for HRs in the Figure S1 footnote

  • Results, lines 217-222 - Figure3 showed no evidence of significant effect of the joint associations of healthy behaviours and SES on mortality from respiratory diseases for ages ≥65 years. Both in males and females it seems that only for groups from 6 to 9 the estimated effects are statistically significant. These evidences should be highlighted in the text.

Response: Thanks, we have highlighted these evidences in the text as shown in page 8 line 245-250: “Figure 3 showed no evidence of significant effect of the joint associations of healthy behaviours and SES on mortality from respiratory diseases for ages ≥65 years. In males and females, for groups including medium SES & no or one healthy behaviour, low SES & four or five healthy behaviours, low SES & two or three healthy behaviours, low SES & no or one healthy behaviour, the estimated effects are statistically significant (in fe-males, group additionally included medium SES & two or three healthy behaviours).”.

  • Discussion, p.8 line 238-239 “ highest risks of respiratory diseases mortality were seen in adults of low SES and no or one healthy behaviours” the finding refers to adults aged ≤65 years

Response: Thanks, we have specified it.

  • Discussion, p.8-9 lines 245-248 the authors emphasized the role of healthy behavior, but this is not evident from the result they are mentioning, i.e. a change in the HR for low SES from 4.48 to 4.44 after adjusting for healthy behaviors. From this specific analysis the impact of healthy behaviors on mortality outcomes according to SES level is minimal. I would say that there is no impact. This part of the text should be revised according to this consideration. The benefits of adopting healthy behaviors (two or more) are evident from the analysis of joint associations with SES levels, as pointed out by the authors in the following part of the discussion.

Response: Thanks for your good suggestions, we have discussed it from analysis of joint associations with SES level as shown in page 9 line 274-298: “Up to now, limited cohort studies reported that the associations of SES with res-piratory diseases mortality. Our study comprehensively explored the associations be-tween SES and respiratory disease mortality. We found that adults of low SES had higher risk of respiratory diseases mortality. One cohort study from Poland reported that higher socioeconomic status was associated with lower risk of mortality due to diseases of the respiratory system which supported our results [8]. Our study acquired death information until 30 June 2020, the onset of the COVID-19 pandemic could have determined an increase in mortality from respiratory diseases staring from March-April 2020. It is should be paid attention that the impact of the pandemic was stronger among the more vulnerable groups of the population (i.e. low educated, low income and unhealthy behavious) that were usually found at higher risk of infection and mortality [26,27]. Therefore, it is important to pay attention to the impact of SES inequalities on the progression of respiratory diseases and take measures to reduce the inequalities of SES or attenuate its harmful effects on respiratory disease mortality us-ing relevant screening and intervention programmes. After adjusted healthy behav-iours, there was a minimal change in the HR for low SES from 4.48 to 4.44. However, we observed the joint associations of healthy behaviours and SES, in same level of SES, the risk of respiratory diseases mortality was lower among adults with more healthy behaviours. The risk of mortality from respiratory diseases was highest among persons of low SES who had no or one healthy behaviours. Healthy behaviours as important influencing factors for health, previous studies reported that it might alleviate the risk of deaths [10-12]. Our results highlight that adherence on healthy behaviours repre-sents potentially modifiable targets for improving the harmful impact of low SES on life expectancy from respiratory disease. The benefits of adopting healthy behaviors (two or more) are evident when considering the joint effect of SES levels and healthy behaviors.”.

  • The onset of the COVID-19 pandemic could have determined an increase in mortality from respiratory diseases staring from march-april 2020. It is known that the impact of the pandemic was stronger among the more vulnerable groups of the population (i.e. low educated) that were usually found at higher risk of infection and mortality. Even if the pandemic affects only a small time frame of the study period and likely do not affect the estimates, I think this aspect should be briefly addressed in the discussion.

Response: Response: Thanks for your good suggestions, we have discussed it from analysis of joint associations with SES level as shown in page 9 line 274-298: “Up to now, limited cohort studies reported that the associations of SES with res-piratory diseases mortality. Our study comprehensively explored the associations be-tween SES and respiratory disease mortality. We found that adults of low SES had higher risk of respiratory diseases mortality. One cohort study from Poland reported that higher socioeconomic status was associated with lower risk of mortality due to diseases of the respiratory system which supported our results [8]. Our study acquired death information until 30 June 2020, the onset of the COVID-19 pandemic could have determined an increase in mortality from respiratory diseases staring from March-April 2020. It is should be paid attention that the impact of the pandemic was stronger among the more vulnerable groups of the population (i.e. low educated, low income and unhealthy behavious) that were usually found at higher risk of infection and mortality [26,27]. Therefore, it is important to pay attention to the impact of SES inequalities on the progression of respiratory diseases and take measures to reduce the inequalities of SES or attenuate its harmful effects on respiratory disease mortality us-ing relevant screening and intervention programmes. After adjusted healthy behav-iours, there was a minimal change in the HR for low SES from 4.48 to 4.44. However, we observed the joint associations of healthy behaviours and SES, in same level of SES, the risk of respiratory diseases mortality was lower among adults with more healthy behaviours. The risk of mortality from respiratory diseases was highest among persons of low SES who had no or one healthy behaviours. Healthy behaviours as important influencing factors for health, previous studies reported that it might alleviate the risk of deaths [10-12]. Our results highlight that adherence on healthy behaviours repre-sents potentially modifiable targets for improving the harmful impact of low SES on life expectancy from respiratory disease. The benefits of adopting healthy behaviors (two or more) are evident when considering the joint effect of SES levels and healthy behaviors.”.

  • The paper need moderate english language revision

Response: Thanks. We have revised the full article.

Round 2

Reviewer 2 Report

All my comments have been answered and I thank the authors for that.

Author Response

Reviewer: 2

All my comments have been answered and I thank the authors for that.

Response: Thanks.

Reviewer 4 Report

The authors have properly addressed to the comments and the paper has greatly improved after the revisions.

There is only one methodological issue that still needs to be clarified:

Lines 151-152 of the revised version, “... end of follow-up (15 November 2021)”. As the outcome of interest for the study is mortality, the date of end of follow-up should be 30 June 2020, i.e. the date until participants are followed-up for vital status and cause of death (paragraph 2.4, lines 134-136)

Author Response

Review 4:

The authors have properly addressed to the comments and the paper has greatly improved after the revisions.

Response: Thanks.

There is only one methodological issue that still needs to be clarified:

Lines 151-152 of the revised version, “... end of follow-up (15 November 2021)”. As the outcome of interest for the study is mortality, the date of end of follow-up should be 30 June 2020, i.e. the date until participants are followed-up for vital status and cause of death (paragraph 2.4, lines 134-136)

Response: Thanks. We have clarified it as “30 June 2020”.